# HashPose: Memory-Efficient Human Pose Estimation via Progressive Hash Codes

## Abstract

Real-time human pose estimation on edge devices demands memory-efficient, high-precision methods. The dominant heatmap approaches, however, scale quadratically with input size, waste computation on background regions, and require slow post-processing. We propose HashPose, a framework replacing heatmaps with progressive hash codes: each keypoint is a binary sequence where successive bits refine localization, cutting memory complexity from $\Theta(HW)$ to $\Theta(log(HW))$. This direct bit prediction avoids dense heatmap-style background computations and removes the need for argmax or non-maximum suppression to decode coordinates. Furthermore, HashPose utilizes image classification backbones without upsampling layers to achieve high accuracy while significantly boosting its speed. We validate HashPose's performance envelope across a wide range of model sizes, ranging from an efficient 3.5M-parameter HashPose-XT model with 0.82 milliseconds frame latency and 85.9% $AP^{50}$ on the COCO, to a 196.8M Large model that achieves a state-of-the-art 91.9% $AP^{50}$ with 5.57 milliseconds frame latency using only the COCO training set. Simultaneously, HashPose has 510x lower output memory than heatmap configurations (0.48 MB vs 244.8 MB) for a typical $256 \times 192$ input, enabling high-throughput pose analysis that maintains high practical precision for on-device applications. Furthermore, its discrete representation is inherently suited for integer-only quantization, offering a clear path to further hardware acceleration on edge devices. Code is provided as supplementary material.

## 1 Introduction

Real-time human pose estimation is critical for mobile applications (e.g., augmented reality, fitness analysis, and robotics Zheng et al. (2025)), but deployment on **resource-constrained** devices remains challenging due to memory and computational limits. Although state-of-the-art approaches achieve high accuracy with heatmap representations (Khirodkar et al., 2024) (Figure 1 (b)), their memory footprint scales quadratically with input resolution during training and inference, rendering high-resolution processing impractical on edge devices. In addition, they apply learning and inference uniformly across all spatial locations, irrespective of background regions that contain no keypoint features. Converting heatmaps to coordinates further adds latency. Regression-based approaches, while avoiding heatmaps, still require densely sampled anchor points that cost memory and computation (Yang et al., 2025; Khanam & Hussain, 2024). In the domain of edge applications, where models must operate under a strict memory and latency budget, we argue that the most meaningful state-of-the-art is the best possible performance achievable within real-world hardware constraints.

Unlike heatmap or regression approaches, we propose **HashPose** - a new framework that encodes each keypoint as a binary sequence in which successive bits narrow the spatial region from body-level precision to joint-level precision (Figure 1 (c)). Our representation, which mirrors the processing of biological vision (Lou & Yu, 2025; Saalmann et al., 2007), achieves provable uniqueness. For an image of size $H \times W$, our $d$-bit codes achieve a worst-case localization error of $\frac{\max(H,W)}{2^{d+1}}$ pixels, with the provably minimal space complexity $\Theta(\log(HW))$ for unambiguous coordinate encoding. The memory scales with the desired precision $\epsilon$ at $\log(\frac{1}{\epsilon})$ - exponentially better than heatmaps at $\frac{1}{\epsilon^2}$.

Building on these foundations, we propose a *direct image2code prediction network* (Figure 3 (a)) generating hash codes from most significant bits (**MSB**) for coarse body regions to least significant bits (**LSB**) for joint-level precision, and introducing minimal latency and memory overhead. In contrast with heatmap's uniform pixel-wise losses (Qu et al., 2022) or regression's monolithic L2 objective (Yang et al., 2025), we design *progressive bit reweighting* (Figure 3 (b)), a self-paced

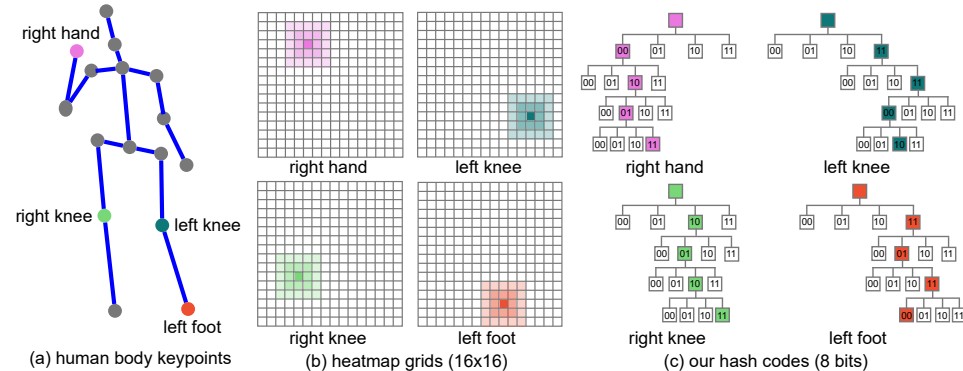

Figure 1: Comparison of (b) typical heatmap grids ($16 \times 16$) with (c) our progressive hash codes (8 bits) for localizing (a) 2D human body keypoints.

technique that adjusts to the varying difficulty of bit prediction across successive levels: early bits (coarse localization) converge rapidly, while later bits (fine positioning) require more nuanced features. Each bit's loss is scaled by its position-dependent weight in the hierarchy and its entropy during training, which effectively progresses learning from coarse to fine and from uncertain to certain.

On top of progressive training, we further propose a *push-pull regularization* (Figure 3 (c)) that eliminates ambiguous "maybe" outputs (e.g., 0.4-0.6 probabilities). The pull component sharpens the prediction of each bit towards definitive 0 or 1 ground-truth values, while the push component penalizes ambiguous predictions that could lead to decoding errors. This dual mechanism effectively enhances the learning of high-precision codes across the spatial hierarchy.

We validate these theoretical advantages across three key areas. We demonstrate significant advances in parameter efficiency, achieving state-of-the-art $AP^{50}$ on COCO with a dramatically smaller prediction head compared to heatmap decoders. The framework also exhibits strong output memory scalability and resolution robustness, showing superior computational scaling up to 4K resolution where traditional methods become impractical. Our main contributions are:

- **Hash Code Representation**: We introduce a progressive hash code representation that guarantees the worst-case localization error at $\frac{\max(H,W)}{2^{d+1}}$ pixels with a provably minimal memory complexity $\Theta(\log(HW))$, improving upon heatmaps and regression.
- **Progressive Code Learning**: We develop (1) a novel image2code network that generates coarse-to-fine bits with improved accuracy, (2) progressive bit reweighting that improves fine-positioning accuracy, and (3) push-pull regularization that reduces prediction errors.
- **State-of-the-Art Efficiency**: The experimental results show 510x output memory reduction (0.48 MB vs. 244.8 MB for $256 \times 192$ inputs), over 2x faster inference (0.82 milliseconds) at least compared to heatmap or regression configurations, and higher accuracy (91.9% $AP^{50}$ on COCO) while supporting 4K resolution at just 0.79 MB output memory.
- **Modular 'Plug-and-Play' Design:** Our framework decouples the hash code prediction head from the feature extractor. This enables the direct use of strong, off-the-shelf image classification backbones without requiring the specialized or computationally expensive upsampling layers common in heatmap-based approaches.

## 2 HASH CODE REPRESENTATION

Here, we propose a hash code representation for pose estimation and formalize its biological basis (Section 2.1), generation algorithm (Section 2.2), and memory-error scaling (Section 2.3).

### 2.1 BIOLOGICAL INSPIRATION FOR CODE DESIGN

The human visual system efficiently refines pose estimates from coarse to fine (Lou & Yu, 2025). Unlike dense heatmaps (Khirodkar et al., 2024) or direct regression (Yang et al., 2025), our hash code design (Section 2.2) emulates key biological efficiencies:

- **Hierarchical Analysis:** Mirroring vision's coarse-to-fine strategy (e.g., Saalmann et al. (2007)), our codes ensure hierarchical spatial containment (C1) and enable early termination of irrelevant pathways via prefix-free pruning (C3).

- **Focused Computation:** Emulating attentional mechanisms that dynamically allocate resources (e.g., Posner (1980) and Itti et al. (2002)), the code structure supports robust prefix-free pruning (C3), rapid constant-time containment checks (C5), and aims for minimal coding redundancy (C4).

## 2.2 HASH CODE GENERATION AND UNIQUENESS

Given an input image $I \in \mathbb{R}^{H \times W \times 3}$, we generate a set of codes of $n$ keypoints, where each keypoint's integer coordinate $p = (x, y) \in [0, W-1] \times [0, H-1]$ on the image plane is mapped to a hash representation $c \in C$, with precision levels $k \in \{1, 2, \cdots, d\}$, where $d = \max(\lceil \log_2 W \rceil, \lceil \log_2 H \rceil)$. The spatial support for code $c$ at level $k$ is denoted as $S(c, k) \subseteq [0, W-1] \times [0, H-1]$. To adhere to the biological principles (Section 2.1), we propose that an efficient hash representation should satisfy some basic constraints:

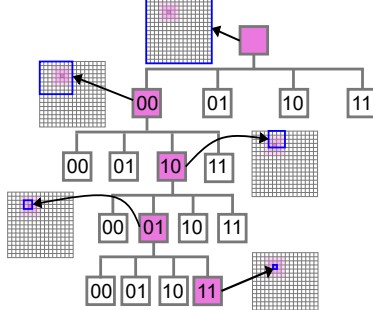

Figure 2: 8-bit binary interleaving code of a keypoint (right hand) on a 16×16 grid. Binary codes for the regions at each level are derived using a 'left/top is 0, right/bottom is 1' assignment rule.

(C1) **Hierarchical Spatial Containment**: The child regions must be the strict subsets of a parent region in the visual processing hierarchy, $S(c, k+1) \subset S(c, k) \, \forall k$.

(C2) **Separable Coordinate Encoding**: The $x$ and $y$ coordinates of keypoints must be independently decodable, $c(x, y) = f(x) \bigoplus g(y)$ where $f(x)$ and $g(y)$ are coordinate-wise encoding functions and $\bigoplus$ is a code composition operator.

(C3) **Prefix-Free Pruning**: The code structure enables efficient identification of distinct spatial hierarchies. For any two codes $c$ and $c'$, if their $k$-th level prefixes differ (i.e., $c_k \neq c'_k$ for some $k \leq d$), then their corresponding sub-hierarchies are distinct.

(C4) **Minimal Coding Redundancy**: There are no unused codes in the $2d$-bit space generated from $d$ precision levels for two dimensions, $|\{c(x, y)\}| = 2^{2d}$.

(C5) **Constant-Time Containment Check**: Verifying the spatial containment of codes $c$ and $c'$ costs minimal time in $\mathcal{O}(1)$.

**Theorem 1** (Representation Uniqueness). *The only representation that satisfies (C1)-(C5) is the binary interleaving encoding $c = [b_1^x, b_2^y, \cdots, b_{2k-1}^x, b_{2k}^y, \cdots, b_{2d-1}^x, b_{2d}^y]$, where for each refinement level $k \in \{1, \ldots, d\}$, $b_{2k-1}^x = \lfloor \frac{x}{2^{d-k}} \rfloor \mod 2$ is the $k$-th binary bit of the $x$-coordinate and $b_{2k}^y = \lfloor \frac{y}{2^{d-k}} \rfloor \mod 2$ is the corresponding bit of the $y$-coordinate (Figure 2).*

*Proof.* (C1) and (C5) together necessitate that the individual coordinate encodings $f(x)$ and $g(y)$, defined by (C2), must be $d$-bit binary strings generated from successive bisection of each coordinate axis; this is required for constant-time containment checks via bitwise operations within a hierarchical structure. (C3) further dictates that these binary strings $f(x)$ and $g(y)$ must order their bits from Most Significant Bit (MSB) to Least Significant Bit (LSB) to enable prefix-based pruning for coarse-to-fine 1D partitioning, establishing them as standard $d$-bit binary representations. For the composed code $c(x, y) = f(x) \oplus g(y)$ (C2) to also satisfy (C1) for 2D regions $S(c, k)$ and (C3) for effective pruning based on 2D prefixes $c_k$, the composition operator $\oplus$ must be interleaving. Interleaving (e.g., $c = [x_1, y_1, x_2, y_2, \ldots]$) ensures that prefixes $c_{2k'}$ (representing $k'$ levels of 2D refinement) correspond to hierarchically nested 2D regions (quad-tree cells). Alternative compositions like concatenation ($c = [x_1, \ldots, x_d, y_1, \ldots, y_d]$) would make initial prefixes $c_{k'}$ (for $k' \leq d$) refine only along one dimension, meaning $S(c, k')$ would not represent progressively smaller, balanced 2D regions suitable for hierarchical 2D pruning as required by (C1) and (C3) jointly. The specific bit extraction formulae in Theorem 1 are the standard definitions for these MSB-first $d$-bit binary representations of $x$ and $y$. The resulting $2d$-bit interleaved code maps each $(x, y)$ pair in the $2^d \times 2^d$ effective grid to a distinct codeword, thereby utilizing all $2^{2d}$ possibilities and satisfying (C4). Thus, the binary interleaving encoding is the unique representation satisfying all constraints (C1-C5). $\square$

## 2.3 Efficient memory-error scaling

With the uniqueness property of our hash code (Section 2.2) in mind, we now demonstrate its theoretical advantages over conventional heatmap and regression approaches.

**Lemma 1** (Worst-Case Localization Error). *For an image of height $H$ and width $W$, the worst-case keypoint localization error is bounded by: (1.1)* **Hash Code** *($d$-bit): $\epsilon_{hash} \leq \frac{max(H,W)}{2^{d+1}}$; (1.2 )***Heatmap** *(stride $s$): $\epsilon_{heatmap} \leq \frac{s}{\sqrt{2}}$; and (1.3)* **Floating Point Regression** *($m$-bit mantissa): $\epsilon_{float} \leq \frac{max(H,W)}{2^{m+1}}$.*

*Proof.* (1.1) For a $d$-bit hash code encoding a coordinate on an axis of effective length $L = \max(H,W)$, the $d$ bits define $2^d$ segments. Each segment has length $\Delta = L/2^d$. The maximum quantization error to the midpoint of a segment is $\epsilon_{hash} \leq \Delta/2 = L/2^{d+1}$. (1.2) For a heatmap where each cell represents an $s \times s$ region of the input, if the decoded coordinate is the cell center, the true coordinate can be anywhere within this cell. The maximum Euclidean distance from the cell center to any corner is $\epsilon_{heatmap} \leq \sqrt{(s/2)^2 + (s/2)^2} = s/\sqrt{2}$. (1.3) For floating-point regression, we assume methods predict normalized coordinates (e.g., $x' \in [0,1]$ for an axis of length $L_x = W$, subsequently scaled to $x = x'L_x$). An $m$-bit mantissa provides an effective precision in the normalized range $[0,1]$ such that the representation error for $x'$ is bounded by $2^{-(m+1)}$ (i.e., half the smallest step effectively representable by the $m$ mantissa bits over the unit range). When scaled by the axis length $L = \max(H,W)$, this yields an absolute localization error $\epsilon_{float} \leq L \cdot 2^{-(m+1)}$. Therefore, the bounds are $\epsilon_{hash} \leq \frac{\max(H,W)}{2^{d+1}}$, $\epsilon_{heatmap} \leq \frac{s}{\sqrt{2}}$, and $\epsilon_{float} \leq \frac{\max(H,W)}{2^{m+1}}$. □

**Lemma 2** (Minimal Space Complexity). *For an image of height $H$ and width $W$, the minimal space complexity required to uniquely and unambiguously encode a keypoint coordinate pair is: (2.1)* **Hash Code**: $\mathcal{M}_{hash} = \Theta(\log(HW))$; (2.2) **Heatmap**: $\mathcal{M}_{heatmap} = \Theta(HW)$; and (2.3) **Floating Point Regression**: $\mathcal{M}_{float} = \Theta(1)$.

*Proof.* A total number of $HW$ unique $(x,y)$ coordinate pairs exist on the image. By Shannon's source coding theorem Wyner (2003), the information-theoretic minimum number of bits required to encode these is $\lceil \log_2(HW) \rceil$. The hierarchical binary partitioning of our hash code generates $\mathcal{M}_{\text{hash}} = 2\lceil \log_2(\max(H,W)) \rceil = \Theta(\log(HW))$ bits for both coordinates, which meets the lower bound and thus makes it minimal. To store a confidence value for each pixel in a downsampled heatmap (stride $s$), we would need $\mathcal{M}_{\text{heatmap}} = (\frac{W}{s} \times \frac{H}{s}) = \Theta(HW)$ values. This makes it exponentially worse than the $\log(HW)$ bound. The number of bits required to store a floating point value is constant irrespective of input size, and therefore also fails to achieve $\log(HW)$ scaling. □

**Corollary 1** (Memory-Error Trade-off). *For a target keypoint worst-case localization error $\epsilon$, the required memory $\mathcal{M}$ for fixed image size $(H, W)$ scales as: (1.1)* **Hash Code**: $\mathcal{M}_{hash} = \Theta(\log(\frac{1}{\epsilon}))$; *(1.2)* **Heatmap**: $\mathcal{M}_{heatmap} = \Theta(\frac{1}{\epsilon^2})$; and (1.3) **Floating Point Regression**: $\mathcal{M}_{float} = \Theta(\log(\frac{1}{\epsilon}))$.

*Proof.* For hash codes, from Lemma 1 (1.1), the error bound $\epsilon \leq \frac{\max(H,W)}{2^{d+1}}$ gives $d \geq \log_2 \frac{\max(H,W)}{\epsilon} - 1$. Since $\mathcal{M}_{hash} = \Theta(d)$ (from Lemma 2 (2.1)), it follows that for fixed $H, W$, $\mathcal{M}_{hash} = \Theta\left(\log\left(\frac{1}{\epsilon}\right)\right)$. For heatmaps, Lemma 1 (1.2) gives $\epsilon \leq \frac{s}{\sqrt{2}}$, implying $s \geq \epsilon\sqrt{2}$. As $\mathcal{M}_{heatmap} = \Theta\left(\frac{HW}{s^2}\right)$ (from Lemma 2 (2.2)), substituting $s$ leads to $\mathcal{M}_{heatmap} = \Theta\left(\frac{HW}{2\epsilon^2}\right)$, which is $\Theta\left(\frac{1}{\epsilon^2}\right)$ for fixed $H, W$. For floating-point regression using a standard type with a $m$-bit mantissa (e.g., IEEE 754 32-bit or 64-bit floats), the memory complexity $\mathcal{M}_{float} = \Theta(1)$ as established in Lemma 2 (2.3). The corresponding worst-case error bound $\epsilon_{float} \leq \frac{\max(H,W)}{2^{m+1}}$ (from Lemma 1 (1.3)) is mainly determined by the $m$ for a given image size. Note that lower-bit model quantization may degrade the prediction accuracy of heatmap and regression methods, but it will not affect 0/1 binary encoding of HashPose. □

## 3 Progressive code learning

After establishing the theoretical advantages (Section 2), we now turn to the practical realization of learning hash codes for pose estimation. The total localization error for a keypoint ($x$ or $y$), $\epsilon_{hash}$,

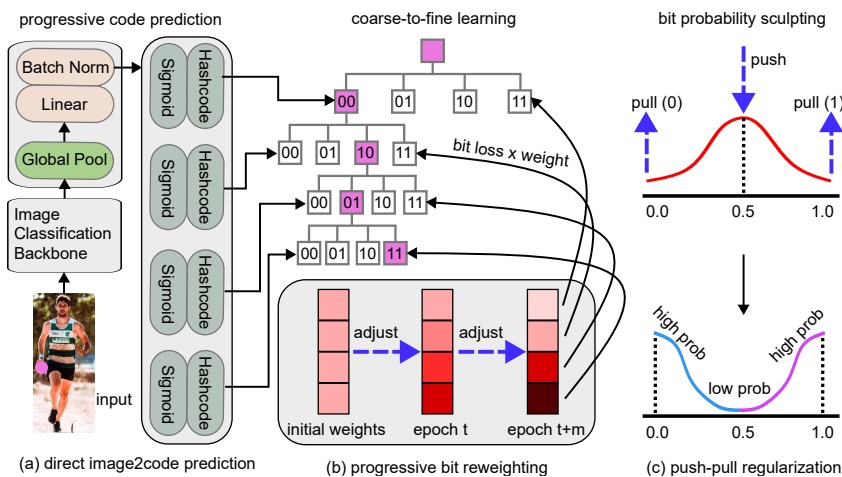

Figure 3: Overview of our hash code learning framework that enables direct image2code prediction (a), progressive code learning (b), and precise bit generation (c).

for a $d$-bit hash code (derived from Lemma 1) is given by:

$$\epsilon_{hash} \leq \sum_{j=1}^{d} \frac{\max(H, W)}{2^{j+1}} \cdot \mathbb{I}(b_j \neq \hat{b}_j) \tag{1}$$

This reveals two key insights: (i) mispredictions in MSBs cause exponentially larger errors than in LSBs, and (ii) MSB errors propagate and cannot be corrected by LSBs due to the prefix nature. These motivate our progressive learning framework: prioritizing MSB accuracy (Section 3.1), adaptively weighting LSB learning (Section 3.2), and ensuring unambiguous bit predictions (Section 3.3).

### 3.1 DIRECT IMAGE2CODE PREDICTION

Our direct image2code prediction network (Figure 3(a)) consists of an image classification backbone and a lightweight code prediction head. Unlike networks that predict heatmaps, the image classification backbone doesn't include upsampling layers. The resulting feature maps $F \in \mathbb{R}^{h \times w \times C}$ (where $h = H/s$, $w = W/s$) have a high spatial compression ratio, typically $s = 8$. This design decouples feature extraction from code prediction, allowing us to use powerful, off-the-shelf image classification models in a *plug-and-play* manner, and significantly reduces the model's parameter count and computational load. The simplicity of this head is a key design feature; it relies on the rich, upstream features from the backbone to handle complex transformations like rotation and deformation, allowing the head to focus solely on the efficient decoding of the final coordinate. The lightweight code prediction head is made up of a global average pooling layer, a linear layer, and a batch normalization layer. The global average pooling layer is used to aggregate a spatial feature map into a vector $F'$. The extracted features $F'$ are then fed into a linear layer, which, along with BatchNorm (BN) for training stability and a sigmoid activation ($\sigma$) for outputting bit probabilities, predicts the $j$-th bits for both x and y coordinates:

$$[b_j^x, b_j^y] = \sigma(\text{BN}(\text{Linear}_j(F'))) \tag{2}$$

### 3.2 PROGRESSIVE BIT REWEIGHTING

To automatically coordinate learning across the predicted bits (Section 3.1), we introduce an adaptive weighting scheme for each bit $j$'s contribution to the loss (Figure 3(b)). The weight $w_j$ is defined as:

$$w_j = (1 + j/d) \cdot (1 + H_j) \tag{3}$$

Here, the term $(1 + j/d)$ linearly increases weight for later bits (LSBs), emphasizing their importance for fine-grained precision, consistent with Lemma 1. The second term, $(1 + H_j)$, uses the binary entropy $H_j = -[p_j \log p_j + (1 - p_j) \log(1 - p_j)]$ of the current predicted probability $p_j$ for bit $j$,

thereby adaptively focusing learning on unstable or uncertain bits. This combination creates a self-paced curriculum, naturally shifting the training focus from initially uncertain MSBs towards more challenging LSBs as learning progresses. This hyperparameter-free reweighting can be efficiently computed and demonstrates robustness across different backbone architectures and datasets.

### 3.3 PUSH-PULL REGULARIZATION

To enhance both geometric accuracy and prediction certainty, we introduce a push-pull regularization loss (Figure 3(c)), $\mathcal{L}_{reg}$, that operates directly on the predicted bits $(b^x_{m,j}, b^y_{m,j})$ for each keypoint $m$. Given a set $\mathcal{P}$ of keypoint coordinate pairs and ground-truth bits $(t^x_{m,j}, t^y_{m,j})$ (derived from annotated joint positions as per Section 2) of a person, the average loss of all keypoints is:

$$\mathcal{L}_{reg} = \frac{1}{2|\mathcal{P}|d} \sum_{m \in \mathcal{P}} \sum_{u \in \{x,y\}} \sum_{j=1}^{d} v_m w_j \left[ BCE(b^u_{m,j}, t^u_{m,j}) + \lambda(0.5 - |b^u_{m,j} - 0.5|)^2 \right] \quad (4)$$

The first term pulls predicts bits $b_{m,j}$ towards their ground-truth values $t_{m,j}$ (subject to the visibility flag $v_m$) using a binary cross-entropy (BCE) loss. The second term, with a balance parameter $\lambda = 0.5$, penalizes ambiguous probabilities around 0.5 by pushing predictions towards definitive 0 or 1 values. This regularization complements the progressive bit reweighting (Section 3.2) by sharpening predictions, especially in challenging cases. The ground-truth bits $t_{m,j}$ are precomputed during data loading with negligible overhead, and $\mathcal{L}_{reg}$ adds minimal computational cost during training.

## 4 EXPERIMENTAL RESULTS

We conduct extensive experiments to validate the HashPose representation framework across a wide range of model sizes on the COCO (Common Objects in Context) (Lin et al., 2014) and MPII Human Pose (Andriluka et al., 2014) benchmarks. We show the framework designs synergize well and provide a strong foundation for high-throughput-high-precision pose analysis. Furthermore, we show that our efficient 3.5M-parameter HashPose-XT model with **0.82** milliseconds frame latency achieves 85.9% $AP^{50}$, our large HashPose-L model can achieve a new state-of-the-art of **91.9%** $AP^{50}$ with 5.57 milliseconds frame latency on the COCO dataset, using only the public training subset and without any test-time data augmentation or post-processing. More details and results are provided in Appendices A-F.

### 4.1 CORE COCO VAL BENCHMARKS

Heatmaps are one of the most widely used representation methods. For typical models with 20 to 90 million parameters, such as RTMPose(Jiang et al., 2023), TokenPose(Li et al., 2021b), SimCC(Li et al., 2022), HRNet(Wang et al., 2021) and ViTPose(Xu et al., 2022), this representation can achieve over 90% AP$^{50}$. However, heatmap-based methods increase the model's parameter count, memory cost, and computational load. They also require additional post-processing, making them unsuitable for resource-constrained devices. Regression and RLE methods (e.g., RLE(Li et al., 2021a) and EDPose(Yang et al., 2025)) directly predict keypoint coordinate values, which saves model parameters and reduces memory consumption. However, RLE requires a complex, auxiliary flow model and carefully tuned priors to estimate coordinate variances during training.

In contrast, the HashPose method significantly reduces memory usage and eliminates the need for post-processing. This is particularly evident when compared to RLE, our HashPose-T model is simpler and delivers superior performance. Specifically, HashPose-T achieves **1.6 points higher** $AP^{50}$ (90.5% vs. 88.9%) with **25% less output memory** (0.48 MB vs. 0.64 MB) than RLE-R50, as shown in Table 1. This demonstrates a clear advantage in both accuracy and efficiency over the state-of-the-art in likelihood-based regression.

Our HashPose models demonstrate significant scalable and speed advantages, as shown in Table 1. For instance, after optimization with TensorRT(Tor) and using bfloat16 precision(Aut), the HashPose-XT model's latency can be reduced to just 0.8 milliseconds with no accuracy loss. We compared HashPose-XT with the lightweight YOLOv11 (Khanam & Hussain, 2024; Jegham et al., 2025) models. HashPose-XT has fewer parameters and less computational load than YOLOv11, but its accuracy is significantly higher, and its per-frame processing latency is also lower. When the parameter count of

Table 1: Comparison of different methods on the COCO-Val dataset. **#P** and **#F** denote the number of parameters and FLOPs of a model, respectively. **#M.** reports the memory cost of model head output. **Xproc.** signifies test-time data augmentation (e.g., flip or multi-scale) and post-processing (e.g., heatmap shifting) to improve model accuracy. H., R., and RLE indicate heatmap, regression, regression with log-likelihood, respectively. HashPose models are measured with an input size of $256 \times 192$. A detailed discussion refers to Section 4.1.

| Type | Method | #P(M) | #F(G) | #M.(MB) | Time (ms) | Xproc. | AP$^{50}$ | AR$^{50}$ |
|------|--------|-------|-------|---------|-----------|--------|-----------|-----------|
|      | Simpl.Res50 | 34.0 | 9.0 | 489.6 | 40(20) | Y(N) | 88.6(88.1) | 92.9(92.6) |
|      | Simpl.Res101 | 53.0 | 12.4 | 489.6 | 46(23) | Y(N) | 89.3(88.7) | 93.4(93.0) |
|      | Simpl.Res152 | 68.7 | 15.8 | 489.6 | 51(25) | Y(N) | 89.3(89.0) | 93.4(93.2) |
|      | RTMPose-L | 29.3 | 4.2 | 142.8 | 8.2 | Y | 89.9 | 93.5 |
| H.   | TokenPose-L | 27.5 | 11.6 | 489.6 | 58.1(29.0) | Y(N) | 90.3(89.5) | 93.7(93.5) |
|      | SimCC-W48 | 66.4 | 15.8 | 142.8 | 57.6(28.3) | Y(N) | 90.4(90.1) | 94.4(94.2) |
|      | HRNet-W32 | 28.6 | 7.7 | 489.6 | 57.8(28.9) | Y(N) | 90.5(90.2) | 94.2(94.1) |
|      | HRNet-W48 | 63.6 | 15.8 | 489.6 | 58.0(29.0) | Y(N) | 90.6(90.0) | 94.1(93.9) |
|      | ViTPose-B | 86 | 17.1 | 489.6 | 67.8 | Y | 90.7 | **94.6** |
| RLE  | RLE-R50 | 23.7 | 3.7 | 0.64 | 52 | Y | 88.9 | 92.6 |
| R.   | EDPose-R50 | 47.9 | 39.2 | 64 | 51 | Y | 89.6 | 94.6 |
| Hash | HashPose-T | 28.0 | 4.4 | **0.48** | **1.7** | **N** | **90.5** | 94.0 |
| R.   | YOLOv11-N | 2.9 | 7.6 | 8.6 | 1.7 | Y | 81.0 | - |
| R.   | YOLOv11-S | 9.9 | 23.2 | 8.6 | 2.6 | Y | 86.3 | - |
| Hash | HashPose-XT | 3.5 | 0.5 | **0.48** | **0.8** | **N** | 85.9 | 90.4 |
| R.   | EDPose-L | 221 | 139.9 | 64 | 88 | Y | 91.5 | 95.0 |
| Hash | HashPose-L | 196.8 | 33.7 | **0.48** | **5.6** | **N** | **91.9** | **95.2** |

HashPose-L is scaled up to 196.8 million, it achieved a new state-of-the-art accuracy, with a per-frame latency of only 5.6 milliseconds. Note that these speeds can be further improved as HashPose's discrete representation is inherently suited for integer-only quantization.

## 4.2 MEMORY USAGE AND SCALABILITY: HASHPOSE VS. HEATMAPS

Figure 4 illustrates HashPose's dramatic advantages in memory usage and scalability over traditional heatmap representations. At a common $256 \times 192$ input, HashPose with bfloat16 precision requires just **0.48 MB** for its output representation compared to **244.8 MB** for heatmaps—a 510-fold reduction. This efficiency is rooted in HashPose's $\Theta(\log(HW))$ complexity versus $\Theta(HW)$ for heatmaps (Lemma 2). Predicting heatmaps requires a more wide and thick backbone model, while HashPose only needs a more slim one. As input sizes increase, heatmap memory consumption grows quadratically, exceeding **3.8 GB** for $1024 \times 768$ inputs. In contrast, HashPose scales exceptionally well, using only **0.64 MB** for the same $1024 \times 768$ resolution and a mere **0.79 MB** for 4K resolution ($4096 \times 3072$), where heatmap methods become practically infeasible for typical hardware.

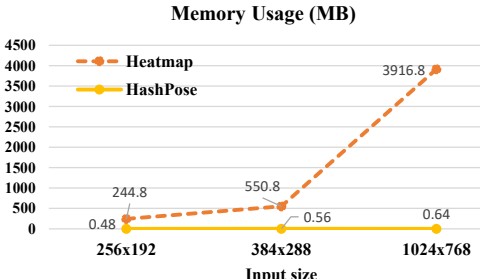

Figure 4: Comparison of memory usage (MB) for HashPose and heatmap representations across varying input image resolutions, illustrating their differing scalability.

## 4.3 ACCURACY PROFILE OF HASH CODES: BIT-WISE AND FINE-GRAINED ANALYSIS

To understand the accuracy characteristics of our hash codes, we first examine per-bit prediction performance. Figure 5 shows that while MSBs, corresponding to coarse spatial localization, are predicted with high accuracy (e.g., >90-95% for initial bit pairs), the accuracy for subsequent LSBs defining finer details progressively decreases, falling to around 60-70% for the 6th bit pair. This trend underscores the inherent challenge in achieving perfect precision for all bits, and directly impacts overall localization at stricter criteria.

Table 2: Comparison of localization accuracy of HashPose and heatmaps using various evaluation thresholds, model and input sizes on the COCO-Val dataset.

| Type/Input | Method | Xproc. | AP | $AP^{50}$ | $AP^{75}$ | AR | $AR^{50}$ | $AR^{75}$ | Time (ms) |
|---|---|---|---|---|---|---|---|---|---|
| H./256*192 | HRNet-W48 | flip+shift | **75.0** | 90.6 | **82.4** | **80.3** | 94.1 | **86.6** | 58 |
| | | w/o | 72.1 | 90.0 | 80.6 | 78.3 | 93.9 | 85.7 | 29 |
| Hash/256*192 | HashPose-L | w/o | 71.2 | **91.9** | 79.9 | 77.3 | **95.2** | 85.3 | **5.6** |
| Hash/256*192 | HashPose-H | w/o | 72.4 | **91.9** | 80.9 | 78.5 | **95.3** | 86.1 | 12.1 |
| H./384*288 | HRNet-W48 | flip+shift | **76.3** | 90.8 | **82.9** | **81.2** | 94.2 | **87.1** | 60 |
| | | w/o | 74.2 | 90.4 | 81.7 | 79.7 | 94.0 | 86.3 | 30 |
| Hash/384*288 | HashPose-L | w/o | 71.8 | **92.0** | 80.6 | 77.8 | **95.3** | 85.9 | **11.1** |

Table 3: Impact of learning components on accuracy (HashPose-L, COCO val, $256 \times 192$ input).

| Bit reweighting | Label smoothing | Push-pull regularization | AP | $AP^{50}$ | $AP^{75}$ |
|---|---|---|---|---|---|
| | | | 67.2 | 91.5 | 75.6 |
| ✓ | | | 67.9 | 91.4 | 76.5 |
| ✓ | ✓ | | 70.4 | 91.9 | 79.3 |
| ✓ | ✓ | ✓ | **71.2** | **91.9** | **79.9** |

This bit-level behavior influences fine-grained localization, particularly when evaluated using $AP^{75}$. As shown in Table 2, while HashPose-L achieves $AP^{50}$ of 91.9% and $AR^{50}$ of 95.2%, exceeding the heatmap representation's 90.0% and 93.9% (both without extra processing), its $AP^{75}$ score of 79.9% is lower than the heatmap's 80.6%. We further use a larger HashPose-H model (658.1M) and find that the $AP^{50}$ remained the same, while the $AP^{75}$ accuracy increased to 80.9%, which is higher than the heatmap's 80.6%. Next, we find that when the models used a larger input ($384 \times 288$), their $AP^{50}$ slightly improved, but their $AP^{75}$ significantly increased. The $AP^{75}$ of HashPose-L increased from 79.9% to 80.6%. This shows that HashPose models with greater capac-

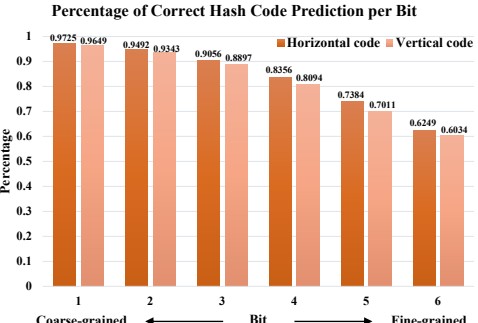

Figure 5: Correct prediction percentage per bit (coarse-to-fine) for horizontal and vertical hash code components (HashPose-Eval, COCO val, $256 \times 192$ input).

ity and larger input are more helpful for improving the fine-grained accuracy. This result is crucial, as it demonstrates that the fine-grained precision of our framework is not fundamentally limited but is instead a tunable property dependent on the backbone capacity. It confirms that HashPose is a scalable representation that can be configured to prioritize either extreme efficiency or state-of-the-art precision.

## 4.4 ABLATION STUDIES

We performed ablation studies on COCO-val to validate design choices, with key results for learning components summarized in Table 3. Our progressive bit reweighting scheme (Section 3.2, Eq. 3), which addresses the varying difficulty of predicting bits (Figure 5), boosts performance by **+0.9%** $AP^{75}$. Applying standard label smoothing (Szegedy et al., 2016) ($\epsilon = 0.2$) to the binary ground-truth bits further significantly improves $AP^{75}$ by **+2.8%** by preventing over-confidence. The proposed push-pull regularization loss ($\mathcal{L}_{reg}$, Eq. 4; Section 3.3) enhances prediction certainty and contributes an additional **+0.6%** $AP^{75}$. We validated our choice of $\lambda$ by grid search from a range between 0.1 and 1.0; $\lambda = 0.5$ provided the best performance. Notably, alternative losses explored, such as contrastive loss or Hamming distance methods (Wang et al., 2023), did not ensure training convergence for our direct hash code prediction. Further investigations (detailed in Appendix E) confirmed that: (i) varying the hash code's effective output resolution (e.g., $64 \times 48$ vs. $256 \times 192$) had minimal impact on $AP^{50}$, unlike changes in heatmap resolution; (ii) an image-independent hashing scheme (Wu & Ghanem, 2019) failed to converge; and (iii) redundant bit prediction (averaging multiple outputs per bit) offered no performance gains. BatchNorm were also found to be crucial for stable training.

## 5 RELATED WORK

**Heatmap Representation.** Predicting body joint locations is challenging due to factors like the narrow profile of extremities, the complex and variable appearance of clothing and backgrounds, and dynamic human movements. Consequently, many current deep neural network methods predict keypoint heatmaps (Li et al., 2021b; Cao et al., 2021; Wei et al., 2016; Khirodkar et al., 2024; Cheng et al., 2020). The concept of heatmap representation for pose estimation was first introduced by Tompson et al. (2014). Typically, deep neural networks output multi-channel heatmaps, where each channel corresponds to a specific joint type, and confidence values on each channel's heatmap approximate the spatial distribution of that joint. For generating these confidence values, one common approach involves selecting a disc-like neighborhood around keypoints, setting values within the disc to 1 and 0 otherwise (Papandreou et al., 2017; He et al., 2017). Another typical method uses a two-dimensional Gaussian distribution, *i.e.,* 1 is assigned at the keypoint location, and the confidence value decreases with distance from the keypoint (Yang et al., 2021; Qu et al., 2022). After heatmaps are inferred, a separate argmax operation is commonly required to retrieve the coordinates of the most confident pixels. Sun et al. (2018) proposed a differentiable soft-argmax operation to enable end-to-end training. SimCC (Li et al., 2022) predicted separate position distribution vectors for horizontal and vertical coordinates, confirming their independent inferability.

**Regression-based Representation.** Regression-based methods directly map an input image to output joint coordinates. However, for scattered keypoints over a large range, deep neural networks often struggle to predict wide-ranging numerical coordinates directly. A common strategy is to define reference or anchor points and then regress relative offsets from these points (Toshev & Szegedy, 2014; Carreira et al., 2016; Wei et al., 2020). For instance, Rogez et al. (2017) discriminates a human pose from a set of anchor-poses and subsequently refines these anchors to their final locations. Li et al. (2021a) boosts prediction accuracy using residual information from keypoint distributions, while Geng et al. (2023) learns a discrete human pose dictionary with a variational autoencoder-based model and then derives the final pose. More recent approaches, like Yang et al. (2025), detect human instances and their keypoints simultaneously, regressing bounding boxes, and often deploy Transformer networks (Shi et al., 2022) to leverage global information for enhanced performance.

**Positioning of HashPose.** Our HashPose representation, in contrast, directly predicts hash codes for all keypoints in an image. Compared to heatmap methods, hash code generation is simpler, and the code length/memory scales logarithmically with input size, addressing critical memory and computational scaling issues. Crucially, this design decouples the task of feature extraction from coordinate representation. While overall performance naturally depends on the backbone's capacity, our core contribution lies in the hyper-efficient head, which dramatically outperforms heatmap and regression-based decoders when compared on similarly powerful backbones. Relative to regression-based approaches, HashPose eliminates the need for predefining reference or anchor points and omits the subsequent computationally intensive discrimination over numerous candidate locations. This unique combination of *simplicity* and *logarithmic scaling* establishes HashPose as a new paradigm for efficient pose estimation.

## 6 CONCLUSION AND FUTURE WORK

We introduced HashPose, a novel keypoint coordinate hashing method enabling direct and memory-efficient keypoint localization. Compared to heatmap and regression techniques, HashPose significantly reduces memory consumption, and eliminates complex post-processing, while achieving a new state-of-the-art $AP^{50}$. It enables efficient pose analysis for memory-limited edge applications. Future work will focus on enhancing fine-grained precision (Section 4.3), potentially via more effective feature learning at fine scales.

While HashPose offers significant memory efficiency, its primary limitation is a slightly lower fine-grained localization accuracy ($AP^{75}$) compared to heatmap methods. This is linked to the challenge of predicting LSBs (Section 4.3), which could be addressed in future work with targeted feature refinement modules. Common to many methods, performance also depends on backbone feature quality, prompting further research into the optimal pairing of backbone architectures and hash code depths. Finally, our current study is limited to 2D pose estimation, though the hash code representation is theoretically extensible to 3D coordinates, presenting a clear path for future investigation.

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

# Appendix:
# HashPose: Memory-Efficient Human Pose Estimation via Progressive Hash Codes

In this Appendix, we provide the descriptions of datasets (in Sec. A), a detailed experimental setup (in Sec. B), memory calculation of heatmap and HashPose (in Sec. C), comparisons under COCO test-dev and MPII (in Sec. D), more ablation study (in Sec. E), as well as a quantitative derivation of OKS thresholds and a qualitative visual analysis (in Sec. F). We disclose the use of large language models (LLMs) for editing assistance of the paper in Sec. G. Our code, pre-trained models, and evaluation tools are provided as supplementary material to ensure reproducibility and facilitate further research.

## A  BENCHMARK DATASETS

We perform qualitative and quantitative experiments on the popular *MPII* human pose dataset Andriluka et al. (2014) and *MS-COCO* keypoint detection dataset Lin et al. (2014). The backbone and prediction head networks are trained using only these two datasets. The MPII dataset contains 24,589 images and 28,883 annotated people. The COCO dataset includes training, validation and testing sets. On the training and validation sets, there are 118,287 and 5000 images respectively, a total of over 150,000 human instances with around 1.7 million labelled keypoints. The testing set has two splits: test-dev and test-challenge, each includes roughly 20,000 images. The proposed model is trained on the training set and evaluated on the validation set. The model is also evaluated on the test-dev set.

In the evaluation, MPII dataset uses the head-normalized probability of correct keypoint ($PCKh@0.5$). COCO keypoint dataset defined a human pose data dedicated overlapping index for matching predictions to groundtruth, object keypoint similarity (OKS), which calculates the overlapping ratio between groundtruth and predictions in terms of point distribution. Based on the OKS index, we report Average Precision (AP), $AP^{50}$ (AP at an OKS threshold of 0.5), $AP^{75}$, and Average Recall (AR), with $AP^{50}$ serving as the primary metric unless specified otherwise.

## B  MODEL AND TRAINING DETAILS

In this paper, human pose estimation follows the two-stage top-down pathway, where a person detector first obtains human regions, then each human instance patch after cropping and scaling operations is input to the model and hash codes are predicted. To facilitate performance comparisons of HashPose with heatmap or regression methods, we follow the previous method Khirodkar et al. (2024) to use the same person detectors. The reported performance results of HashPose models are based on the pure convolutional backbone of ConvNeXt-v2 model family (Woo et al., 2023), which downsamples feature maps to 1/8 input resolution. The model statistics of HashPose are shown in Table 4.

Table 4: Model statistics of HashPose for the same input size of $256 \times 192$.

| HashPose | #P(M) | #F(G) | AP$^{50}$ | Time (ms) |
|:---:|:---:|:---:|:---:|:---:|
| XT | 3.5 | 0.5 | 85.9 | 0.8 |
| T | 28.0 | 4.4 | 90.5 | 1.7 |
| L | 196.8 | 33.7 | 91.9 | 5.6 |
| H | 658.1 | 112.8 | 91.9 | 12.1 |

We train HashPose for 300 epochs using AdamW (Loshchilov & Hutter, 2019) with a learning rate of 0.0018. We use a 20-epoch linear warmup and a cosine decaying schedule afterward, an effective batch size of 1024 via gradient accumulation, a weight decay of 0.05, and employing bfloat16 mixed precision and Channel Last memory layout for speedup. The random data augmentation is analogous to the steps in HRNet Wang et al. (2021). The invisible or unlabeled keypoints are assigned the hash

Table 5: Comparisons on the COCO test-dev set.

| Approach | #P(M) | #F(G) | Time(ms) | Xproc. | AP$^{50}$ |
|---|---|---|---|---|---|
| ResNet-152 | 68.6M | 38.3 | 53 | Y | 91.9 |
| HRNet-W48 | 63.6M | 35.5 | 62 | Y | 92.5 |
| TokenPose-L/D24 | 27.5M | 11.6 | 58.1 | Y | 92.1 |
| SimCC-W48 | 66.4M | 15.8 | 57.6 | Y | 92.4 |
| ViTPose-B | 86M | 17.1 | 67.8 | Y | 92.5 |
| ED-Pose | 221M | 139.9 | 88 | Y | 92.3 |
| HashPose-L | 196.8M | 33.76 | 5.6 | N | **93.0** |

Table 6: Results on the MPII validation set (PCKh). The input size of HashPose is 256×256.

| Method | #P(M)/#F(G)/T(ms) | Hea. | Sho. | Elb. | Wri. | Hip. | Kne. | Ank. | Avg |
|---|---|---|---|---|---|---|---|---|---|
| ResNet-152 | 68.6 / 22.7 / 54 | 97.0 | 95.9 | 90.0 | 85.0 | 89.2 | 85.3 | 81.3 | 89.6 |
| HRNet-W32 | 28.5 / 10.2 / 59 | 96.9 | 95.9 | 90.5 | 85.9 | 89.1 | 86.1 | 82.5 | 90.0 |
| HRNet-W48 | 63.6 / 21.1 / 63 | 97.2 | 95.7 | 90.7 | 85.6 | 89.0 | 86.8 | 82.2 | 90.1 |
| TokenPose-D6 | 21.4 / 12.8 / 53 | **97.1** | 95.9 | **91.0** | 85.8 | **89.5** | 86.1 | 82.7 | 90.2 |
| TokenPose-D24 | 28.1 / 14.6 / 59 | **97.1** | 95.9 | 90.4 | **86.0** | 89.3 | **87.1** | 82.5 | 90.2 |
| HashPose-T | 28.0 / 5.9 / **1.9** | 97.0 | 95.9 | 90.7 | 85.9 | **89.5** | 86.7 | **83.2** | **90.3** |

codes of all 0s. During testing, the predicted hash codes of all 0s will not be further transformed into coordinates. The models are trained on NVIDIA H20 GPUs and tested on a RTX 3090 GPU.

## C   MEMORY CALCULATION FOR HASHPOSE AND HEATMAPS

Compared to heatmap representation, an advantage of HashPose is that the number of parameters required for hash codes is greatly decreased. For an input image of size $H \times W$, the heatmap size is often reduced to $H/4 \times W/4$. The number of keypoint types is $n = 17$ on the COCO dataset. Heatmaps and hash codes are expressed in the format of 32-bit and 16-bit floating point, respectively. The memory usage (MB) of the heatmap representation can be calculated as follows:

$$M = \frac{(B \times n \times H/4 \times W/4) \times 4 \times 1.2}{1024} \tag{1}$$

where $B$ is batch size, 4 expresses 4 bytes used for each parameter, 1.2 represents a 20% overhead of loading additional things in GPU memory. During training, the memory usage doubles because backpropagation gradient values also need to be stored.

The memory usage (MB) of HashPose can be computed as:

$$M = \frac{B \times n \times (\lceil \log_2(\frac{H}{4}) \rceil + \lceil \log_2(\frac{W}{4}) \rceil) \times 2 \times 1.2}{1024} \tag{2}$$

where $\lceil * \rceil$ is a ceiling function that rounds the number to upper integer.

## D   MORE COMPARISONS ON COCO TEST-DEV AND MPII

To further validate our approach on the official blind test set, we submitted HashPose-L for evaluation on the COCO test-dev server. As shown in Table 5, our method achieves a state-of-the-art $AP^{50}$ of 93.0%. This result surpasses prominent and highly-tuned methods including HRNet-W48 (92.5%), ViTPose-B (92.5%), and ED-Pose (92.3%). Table 6 also shows the improvement of HashPose on the MPII validation set. Achieving the top scores on these datasets confirms the robustness of our hash code representation and establishes HashPose as a leading method for high-performance pose estimation.

# E   ADDITIONAL ABLATION STUDIES

**Ablation on Effective Output Resolution.** The number of bits in a hash code represents a potential numerical range. Heatmap-based methods commonly downsample heatmaps to 1/4 input resolution to achieve a trade-off between prediction accuracy and computation cost. Here we would like to check the best ratio of output to input resolution for HashPose. For an input size $256\times192$, we adapt superparameters of HashPose and map hash codes to various output resolutions $64\times48$, $128\times96$, $256\times192$, and $512\times384$, respectively. The prediction accuracies of models with these output resolutions are shown in Table 7, and 0.5 is the best ratio. We select ratio=0.5 as the default parameter for all HashPose variants.

Table 7: The ratio of output to input resolution of HashPose-T analysis.

| Ratio | 0.25 | 0.5 | 1 | 2 |
|---|---|---|---|---|
| $AP^{50}$ | 90.2 | **90.5** | 90.5 | 90.5 |

**Various Hash Algorithm.** Wu & Ghanem (2019) introduces a hash algorithm, where each hash code is like a multi-hot vector, and each pair of codes has a predefined minimal Hamming distance. We implemented it and assigned the codes to potential integer coordinates. The codes of keypoints are selected as learning targets. However, the model cannot converge during training. This phenomenon indicates that the hashing process which is independent of image may not be suitable for keypoint localization.

**Redundant Prediction.** Redundant prediction means inferring more bits than required length. For example, the head network predicts 3 outputs for every code. Then these 3 outputs are averaged or the most confident one is selected as the last result. The comparison experiments show that the accuracies of standard and redundant prediction are totally the same. Redundant prediction cannot provide any benefit.

# F   QUALITATIVE RESULTS

## F.1   DERIVATION OF PIXEL ERROR FROM OKS THRESHOLDS

To provide a quantitative foundation for the "sub-4-pixel deviation" mentioned in Section 4.3, we can analyze the relationship between the Object Keypoint Similarity (OKS) metric and the localization error in pixel space. The OKS is defined as $\exp(-d^2/2s^2k^2)$, where $d$ is the Euclidean distance between the predicted keypoint and the ground truth, $s$ is the scale of the object, and $k$ is a per-keypoint constant. For a typical keypoint like an ankle on a medium-sized person in COCO, we can solve for the pixel distance $d$ that corresponds to a given OKS threshold.

- At the $AP^{50}$ threshold (OKS = 0.50), the acceptable pixel error $d$ is approximately 5-6 pixels.

- At the much stricter $AP^{75}$ threshold (OKS = 0.75), the acceptable pixel error $d$ shrinks to approximately 2-3 pixels.

This calculation confirms that the drop in performance at $AP^{75}$ is due to the model failing to meet an extremely strict localization requirement of just a few pixels, a level of precision that may be unnecessary for many real-time applications.

## F.2   QUALITATIVE ANALYSIS OF THE PRECISION-EFFICIENCY TRADE-OFF

To qualitatively contextualize this numerical gap, we provide a detailed visual analysis in the Appendix (see Figure 6). This analysis reveals that the 0.7% difference in $AP^{75}$ between our HashPose-L and the heatmap baseline HRNet-W48 (shown in Table 2) without extra processing often corresponds to a sub-4-pixel deviation. We argue that this is a functionally negligible trade-off for the significant 510x memory savings in many real-world, resource-constrained applications.

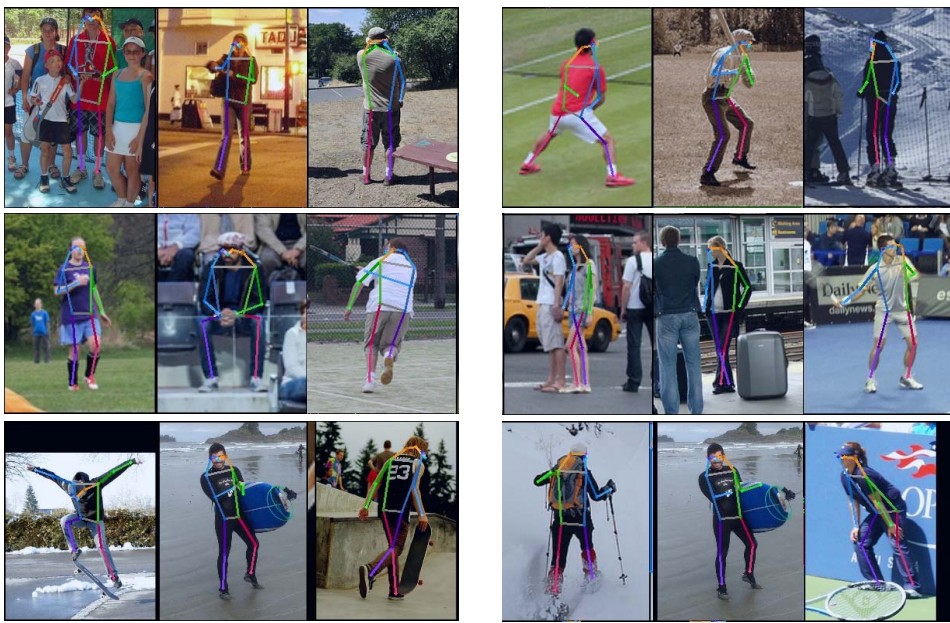

Figure 6: **Visualizing the Precision-Efficiency Trade-off.** A side-by-side comparison of a keypoint (e.g., an ankle) predicted by HRNet-W48 (left) and our HashPose-L (right). The pixel deviation between the two predictions is minimal and often functionally irrelevant for real-time applications like fitness tracking. This small trade-off in high-frequency precision is the source of the 0.7% difference in $AP^{75}$ and is made in exchange for a **510x reduction in output memory**, highlighting the practical value of our method.

To provide a practical context for the fine-grained precision metrics discussed in Section 4.3, Figure 6 presents a qualitative comparison between our HashPose-L model and the high-precision HRNet-W48 heatmap baseline.

# G  LLM USAGE

Our core method development does not involve LLMs as any important, original, or non-standard components. Any use of LLMs was restricted to assistance with writing, editing, or formatting.

