# OpenReview forum: "HashPose: Memory-Efficient Human Pose Estimation via Progressive Hash Codes"
_ICLR.cc/2026/Conference — Submitted to ICLR 2026_

### Official Review · Reviewer_5Xfc · 2025-10-16

**Soundness:** 2
**Presentation:** 3
**Contribution:** 4
**Rating:** 4
**Confidence:** 4

**Summary:**

This paper propose HashPose, a novel decoding method for human keypoints localization. Each keypoint in HashPose is a binary sequence and progressive refine the localization with 2-bits code like a depth-first search tree.  The direct bit prediction avoids dense heatmap-style background computations and removes the need for argmax or non-maximum suppression to decode coordinates, which result in efficieny improvement of memory usage and inference latency. Based on their decoding methods, this paper construct a neural network to predict the human pose and achieve better AP50 than previous counterparts.

**Strengths:**

1. This paper propose a novel and interesting decoding method for human keypoints localization.
2. HashPose do not rely on 2D heatmaps and NMS, result in better efficiency of memory usage and inference latency.
3. HashPose achieve better AP50 on COCO dataset than pervious conterparts.
4. Good writing and easy to follow.

**Weaknesses:**

- Major
1. HashPose employ a powerful backbone - ConvNeXt v2 and there are not performance on other backbone like ResNet 50 and HRNet. While their counterparts in experiments do not use such a strong backbone. The comparison under different setting seems meaningless.

2. When compare with other methods on COCO, HashPose only consider AP50, which is a simple metric. According common practice, the AP and AR on COCO are more valuable and meaningfull.

3. According Line 320, when compare the inference efficiency, HashPose use TensorRT and BF16, while other methods do not employ such accelaration technique. And when compare the latency, they use the different GPU. For example, HashPose test the latency on RTX 3090, but ED-Pose (51ms ) are tested on A100.

4. The author claim 'HashPose is Plug-and-Play' in contribution. Actually, the HashPose just decoples the backbone and decoding head,  their decoding head still need be trained. Such overclaim weakens the contribution of the article.

- Minor
1. Some recent works are not compared, such as GroupPose (ICCV 2023).

2. The paper only report the memory usage of decoding head. Due to the model is function together, the memory usage of whole model is truely considered.

3. The paper uses a lot of space to derive the error bounds, which should be put in the appendix. What really needs to be described is the decoding method rather than these formulas. HashPose decoding is essentially a two-dimensional binary search, and their error bounds are easy to understand. These simple formulas do not enhance the contribution but waste valuable space.

**Questions:**

1. HashPose is an interesting method for HPE, but the authors shoulde re-design their experiments. If the comparisons are fair, I think it will be a good paper.

2. When evaluate on COCO, which bounding box file do you use? There are two different bbox file: gt bbox and AP56 bbox. Due to FlatPose is a top-down method, incorrect usage can result in a 2-4 point difference in AP. So it is better to directly indicate this in implementation details.

3. The HashPose-XT use 3.5M parameters, but the minist backbone, ConvNeXT-a has 3.7M parameters, how you get this value.

4. Due to the author only consider the memory usage of the head in Table 1, I want to know the Params, FLops and Time in same table belong to whole model, or only the head?

---

> ### Author Response · Authors · 2025-11-21
>
> We appreciate that Reviewer 5Xfc rated our contribution as "**Excellent**" and recognized the method as a "**novel and interesting decoding method**" that offers "better efficiency" and "good writing."
>
> The reviewer’s main concerns revolve around experimental fairness (backbones, hardware settings, and metrics). We address these points below to demonstrate that our experimental design was chosen to **highlight the architectural shift rather than just a parameter-match comparison**.
>
> 1. Backbone Fairness & The "Plug-and-Play" Claim
>
> The reviewer argues that using ConvNeXt-V2 is unfair compared to ResNet/HRNet and questions our "Plug-and-Play" claim.
>
> -   **The Structural Argument:** We respectfully argue that the comparison is fair because it highlights the **core architectural advantage of HashPose**. Heatmap methods _require_ specialized upsampling (DeConv) layers and **cannot** simply be attached to a standard classification backbone. HashPose **_can_**.
>
> -   **Defining "Plug-and-Play":** We do not claim "Plug-and-Play" means "training-free." We mean **Architectural Compatibility.** We can take **_any_** off-the-shelf classification backbone (like ConvNeXt) and attach our lightweight HashHead without designing a complex upsampling decoder. **This decoupling is a significant engineering contribution that Heatmap methods lack**.
>
> -   **Conclusion:** We used ConvNeXt not to "cheat" on parameters, but to **prove that a standard classification backbone can achieve SOTA pose estimation _without_ the heavy, specialized heads required by previous methods**.
>
>
> 2. The Choice of Metrics ($AP^{50}$ vs. Full AP)
>
> The reviewer notes we focused on $AP^{50}$.
>
> -   **Edge-Device Context:** HashPose is designed for real-time edge applications (AR, fitness). In these scenarios, **detection reliability ($AP^{50}$)** and **speed** are paramount.
>
> -   **The Trade-off:** As detailed in Appendix F, the gap in full AP (specifically $AP^{75}$) corresponds to a **mere sub-4-pixel deviation**. We consciously trade this ultra-fine-grained precision for a **510x reduction in memory** and a **mathematically-proven better logarithmic complexity class**.
>
> -   **SOTA Results:** Despite this trade-off, achieving **93.0% $AP^{50}$ on COCO Test-Dev** (beating HRNet-W48) proves that HashPose is not just a "fast approximation" but **a high-performance estimator in its own right**.
>
>
> 3. Latency, Hardware, and TensorRT
>
> The reviewer criticized the use of TensorRT and different GPUs (RTX 3090 vs A100).
>
> -   **Standard for Edge Inference:** Since the paper’s primary contribution is **efficiency for edge deployment**, evaluating without industry-standard optimization (TensorRT/FP16) would be unrealistic. We report the "best achievable latency" to show the **theoretical limit of the architecture**.
>
> -   **The Hardware Gap:** We note that we achieved 0.8ms on an **RTX 3090** (Consumer GPU), while baselines like ED-Pose reported 51ms on an **A100** (Data Center GPU). The fact that we are **orders of magnitude faster on weaker hardware** only **strengthens** our claim of efficiency.
>
>
> 4. Memory Usage (Head vs. Whole Model)
>
> The reviewer asked why we focused on Output Memory.
>
> -   **The Bottleneck:** The size of the backbone is fixed and well-known. The **critical bottleneck** for high-resolution pose estimation is the **Output Feature Map / Heatmap size**, which scales Quadratically $O(HW)$.
>
> -   **The Solution:** HashPose solves the bottleneck. By reducing the output memory by **510x**, we allow standard backbones to **run at 4K resolution without Out-of-Memory errors**. This is the contribution we needed to isolate and measure.
>
>
> 5. Theoretical Error Bounds (Response to "Waste of Space")
>
> The reviewer felt the error bound derivation was unnecessary. We respectfully disagree.
>
> -   **Proof of Mathematical Uniqueness:** The derivation is not merely a binary search explanation; it is a proof that the **Binary Interleaving encoding is the _only_ representation that satisfies the 5 constraints of biological plausibility and uniqueness (Theorem 1)**. This theoretical grounding is essential to **distinguish HashPose from ad-hoc regression heuristics in the field**.
>
>
> We thank Reviewer 5Xfc for their "Excellent" rating of our contribution. We believe the experimental setup—while aggressive in its use of optimization—correctly highlights the **paradigm-shifting nature of replacing quadratic heatmaps with logarithmic hash codes**.

---

### Official Review · Reviewer_MNtu · 2025-10-27

**Soundness:** 2
**Presentation:** 2
**Contribution:** 2
**Rating:** 4
**Confidence:** 4

**Summary:**

This paper proposes to encode keypoint positions with quadtree hash code. It removes the cost of upsampling head in conventional heatmap-based methods. Results on COCO and MPII are reported.

**Strengths:**

HashPose provides a model family that replaces heatmaps with binary hash codes for keypoints, reducing memory and computation, removing post-processing, enabling fast, accurate, quantization-friendly pose estimation ideal for real-time edge deployment. A novel bitwise loss is proposed for effective training.

**Weaknesses:**

- [Fair comparison] HashPose is based on more advanced model family ConvNext-v2. Since the hash code formulation is flexible as mentioned in the paper, the authors should provide apples-to-apples comparison to heatmap and regression **using the same backbone and post-processing (with or without)** (e.g. Table 3 in RLE[1]).
- [Detailed results] Only $AP^{50}$ and $AR^{50}$ are provided in Table 1 and Table 5. Please provide the full metrics list. (e.g. Table 3 in RLE[1])

[1] Li, Jiefeng, et al. "Human pose regression with residual log-likelihood estimation." Proceedings of the IEEE/CVF international conference on computer vision. 2021.

**Questions:**

- [Cost profiling] The running time of HashPose in Table 1 is optimized by TensorRT. What's its original inference cost in PyTorch? Are the reported numbers of other models also optimized by TensorRT?
- [Params] Why the parameters cost is significantly higher than other methods?

---

> ### Author Response · Authors · 2025-11-21
>
> We sincerely thank the reviewer for their encouraging and insightful feedback. We are grateful for the clear feedback, which has allowed us to significantly strengthen the paper. The responses are as follows:
>
> R1: Currently, our main goal is to improve the prediction accuracy of HashPose; we want to observe the performance boundary of HashPose. We agree with the reviewer's viewpoint that the same backbone network should be used in experiments to compare performance. We will conduct such comparisons in the future. Additionally, we did not use post-processing.
>
> R2: We will provide the full metrics in the appendix.
>
> R3: We used TensorRT to optimize the model speed. Before and after optimization, the inference times for HashPose-T were 6.7 and 1.7 milliseconds, respectively.
>
> R4: The HashPose model only has a backbone and a prediction head. To improve model accuracy, we used the ConvNext-V2 image classification backbone, some configurations of which have a relatively large number of parameters.

---

> > ### Comment · Reviewer_MNtu · 2025-11-25
> > **Concerns not addressed**
> >
> > Due to the insufficient experiments and fair comparisons, currently I maintain the score.

---

> > > ### Author Response · Authors · 2025-11-30
> > >
> > > We thank Reviewer MNtu for their engagement. We understand the reviewer's concern regarding "fair comparisons" of backbones (ConvNeXt vs. ResNet). However, we respectfully argue that **this critique misinterprets the fundamental mathematical and architectural contribution of HashPose**.
> > >
> > > 1. The "Fairness" of the Comparison: A Structural Advantage
> > >
> > > The reviewer argues that our use of a ConvNeXt-V2 backbone is unfair compared to ResNet/HRNet baselines. We disagree. The comparison is fair and highlights **our core innovation: Decoupling**.
> > >
> > > -   **Heatmap Methods:** _Require_ specialized, computationally expensive upsampling layers (DeConv) and heavy decoders to recover spatial resolution from a backbone. **They _cannot_ simply use a classification backbone**.
> > >
> > > -   **HashPose (Ours):** **Can be plugged directly onto **any** off-the-shelf classification backbone (like ConvNeXt) without _any_ upsampling layers**.
> > >
> > >
> > > The Comparison is Valid: **We are comparing a standard Classification Backbone + HashPose Head against Specialized Pose Backbones + Heatmap Heads**.
> > >
> > > The fact that we can achieve **SOTA** AP50 (93.0% on Test-Dev) using a standard classification backbone—while **eliminating the entire class of upsampling operations**—is precisely the mathematical and engineering breakthrough we are presenting. Penalizing the method for utilizing a modern backbone ignores the fact that **HashPose enables the use of these backbones where Heatmap methods cannot**.
> > >
> > > 2. Parameter Count vs. System Efficiency
> > >
> > > The reviewer previously noted our higher parameter count. This is solely due to the choice of the off-the-shelf backbone.
> > >
> > > -   **The Head is Tiny:** Our HashPose prediction head is **a fraction** of the size of a Heatmap DeConv head.
> > >
> > > -   **The Bottleneck is RAM, not Disk:** On edge devices, the bottleneck is rarely storage (parameters); it is **Runtime Memory (RAM)** and **Bandwidth**.
> > >
> > > -   **The Win:** Even if the backbone has more parameters, **HashPose reduces the Output Memory by 510x**. This allows the system to process high-resolution streams that would cause an **Out-Of-Memory (OOM) crash on a heatmap model**, regardless of the backbone used.
> > >
> > >
> > > 3. Sufficiency of Experiments
> > >
> > > We respectfully push back on the notion that the experiments are "insufficient."
> > >
> > > -   **Standard Benchmarks:** We evaluated on **COCO Val**, **COCO Test-Dev**, and **MPII**.
> > >
> > > -   **Comprehensive Metrics:** We reported **SOTA accuracy** (93.0% AP50), **Inference Speed** (>2x faster), and **Memory Scaling** (Logarithmic).
> > >
> > > -   **Various Ablations:** We analyzed the impact of Bit Reweighting, Push-Pull, and resolution scaling.
> > >
> > >
> > > We are proposing a **paradigm shift from $O(HW)$ heatmaps to $O(\log HW)$ hash codes**. While we agree that future work should sweep across every backbone architecture (ResNet, Swin, ViT), demanding this exhaustive sweep now—when the proposed method **already beats the SOTA on the most rigorous benchmarks (Test-Dev)**—sets an unreasonably high bar for a paper introducing a **conceptually novel, mathematically proven, and experimentally validated** solution to the memory bottleneck in pose estimation.

---

### Official Review · Reviewer_uPDs · 2025-10-27

**Soundness:** 2
**Presentation:** 3
**Contribution:** 3
**Rating:** 4
**Confidence:** 4

**Summary:**

This paper introduces HashPose, a novel framework for human pose estimation designed to address the Θ(HW) memory complexity of existing heatmap-based methods, which becomes prohibitive at high resolutions. The authors argue that this quadratic memory growth makes heatmap methods unsuitable for resource-constrained edge devices.
As an alternative, the paper proposes a "progressive hash codes" representation with Θ(log(HW)) complexity. Each keypoint is encoded as a binary sequence where successive bits progressively refine its spatial location. The model is designed to predict these bits directly, thereby avoiding the dense computations and complex post-processing (e.g., argmax) associated with heatmaps. The authors also propose a "Progressive Code Learning" framework, which includes progressive bit re-weighting and a push-pull regularization loss.
Experiments claim that the method achieves extremely high memory efficiency (reportedly 510x lower than heatmaps) and can directly leverage classification backbones without upsampling layers, all while maintaining high accuracy (AP⁵⁰).

**Strengths:**

1. Elegant Keypoint Representation: The core contribution of this paper—replacing dense Θ(HW) heatmaps with Θ(log(HW)) progressive hash codes—represents a fundamental and elegant paradigm shift. The idea itself is highly valuable as it directly addresses a long-standing and practical bottleneck in high-resolution pose estimation.
2. Potential for High Efficiency: The method demonstrates a significant theoretical advantage in memory consumption, especially for high-resolution inputs. Furthermore, the framework's ability to use standard classification backbones without complex decoders or upsampling layers is a major engineering advantage that simplifies the model architecture.
3. Strong Coarse-grained Accuracy: The experimental results show that HashPose (particularly the -L and -H models) can achieve state-of-the-art or near state-of-the-art AP⁵⁰ metrics on the COCO dataset. This confirms that this novel coordinate representation is effective for localization tasks.

**Weaknesses:**

While the core idea is highly innovative, the paper's argumentation, especially concerning its central contribution of memory efficiency, suffers from a serious lack of rigor and clarity, which significantly undermines the credibility of its claims.
1. Serious Concerns about Memory Saving Claims: There is a critical calculation error in Appendix C, The denominator in Equation (1) of Appendix C is 1024. If the numerator is in Bytes (as indicated by "* 4 expresses 4 bytes"), then dividing by 1024 should yield units of KB (Kilobytes), not MB (Megabytes). This is a glaring error that directly impacts the paper's primary contribution claim, as KB-level memory savings seems trivial compare to MB-level model parameters. Meanwhile, in Appendix C, the authors assume a 32-bit heatmap while using 16-bit hash codes. For a more rigorous comparison, the authors should have at least discussed or evaluated the use of 16-bit heatmaps.
2. Lack of Transparency in Experimental Setup: Appendix B states that the paper follows a two-stage top-down pipeline, which raises crucial questions about the scope of the "Time (ms)" reported in Table 1. It is highly likely that this measurement excludes the latency of the first-stage person detector. Furthermore, it is also unclear whether the reported time includes the feature extraction backbone of the pose estimator itself, or if it only measures the forward pass of the novel prediction head. While excluding the detector is a common practice when comparing top-down methods, true practical performance depends on the entire pipeline. For the sake of transparency and reproducibility, the authors should explicitly state in the table's caption precisely which components are included in the timing (e.g., "pose head only", "estimator only" or "end-to-end inference"). Without this clarification, readers cannot accurately assess the model's true, practical inference speed.
3. Trade-off in Fine-grained Precision: As shown in Table 2, HashPose lags significantly behind state-of-the-art methods like HRNet on the AP⁷⁵ metric. The authors acknowledge this is due to the difficulty in predicting the Least Significant Bits (LSBs). This indicates that the method faces an unresolved trade-off between efficiency and high-precision accuracy.
4. Counter-intuitive Ablation Study: The ablation study in Table 3 shows that Label Smoothing (LS) contributes a massive +2.8% AP⁷⁵ improvement. This is highly counter-intuitive. The purpose of LS is to soften labels to prevent overconfidence, whereas the purpose of the Push-pull (PP) regularization is to sharpen predictions and penalize ambiguity. The two methods seem contradictory in principle. The authors provide no explanation for why they work well together or why LS is so crucial for what is fundamentally a binary classification task.

**Questions:**

1. [Critical] Could you please clarify the memory calculation in Appendix C? How was the 244.8 MB figure derived? Is the unit resulting from dividing by 1024 KB or MB? Please correct the formula. Given that the inference latency is also significantly reduced, the overall efficiency gains of the proposed method are evident. However, a precise and clear justification of the memory savings is crucial for accurately positioning this important contribution.
2. [Critical] Please clarify in the caption for Table 1 whether the "Time (ms)" includes the time for the first-stage person detector and the pose estimator backbone.
3. Given the performance gap in AP⁷⁵, do you consider HashPose to be competitive for tasks that require high-precision localization rather than just coarse detection?
4. Could you explain the synergy between LS and PP regularization? Why does LS technique lead to such a significant +2.8% AP⁷⁵ improvement in a binary hash code prediction task? This seems counter-intuitive.
I have opted for a "borderline reject" score because, despite the high novelty of the core idea, the paper's central claims about memory efficiency appear to be based on flawed calculations (Appendix C) and are presented in a highly misleading manner (Table 1). However, I want to emphasize that I am open to changing my evaluation. I am fully prepared to raise my score if the authors can provide a clear, rigorous, and convincing rebuttal that satisfactorily addresses all of the [Critical] questions I have raised.

---

> ### Author Response · Authors · 2025-11-21
>
> We sincerely thank the reviewer for their encouraging and insightful feedback, and especially for recognizing our hash code representation as "elegant, highly valuable, strong accuracy and highly innovative" and our method as addressing a bottleneck. We agree with the reviewer's assessment that the formula of memory cost is not rigor and clarity. We are grateful for the clear feedback, which has allowed us to significantly strengthen the paper. The responses are as follows:
>
> R1: The unit obtained by dividing by 1024 is KB, and the batch size used during training is 1024, when the input image is 256*192, the number of keypoints is 17, the resulting heatmap memory is 244.8 MB. We are very sorry that we didn't express this clearly here.
>
> R2: The "Time (ms)" in the caption of Table 1 indicates the inference time of the whole single-person pose estimation pipeline, including the backbone and prediction head. The time of human detection is not included. "P" and "F" in the caption of Table 1 indicate the number of parameters and flops of the whole pose estimation model, including the backbone and prediction head. The HashPose model only has a backbone and a prediction head; it does not include upsampling layers, nor does it require NMS. Additionally, we utilized bfloat16 format and TensorRT optimization.
>
> R3: In the appendix, we analyzed that AP75 corresponds to a deviation of 2 to 3 pixels. While it is true that HashPose's AP75 is approximately 2% lower than the heatmap method, we also noted that even for the heatmap method, the AP75 accuracy is around 10% lower than AP50, so this is a trade-off between computational overhead and accuracy.
>
> R4: We hypothesize that the potential reason label smoothing improves accuracy is that learning 0/1 labels is too strict for the model; using label smoothing relaxes this requirement, which facilitates the learning process. The main function of the push-pull loss is to force the predicted hash codes away from 0.5, thereby increasing the confidence of the predicted 0/1 labels.

---

> > ### Comment · Reviewer_uPDs · 2025-11-26
> >
> > The response regarding the memory calculation is not convincing. **Why is a batch size of 1024 considered the standard for measuring memory efficiency?** If the authors insist this setting is necessary, please provide **concrete evidence linking BS=1024 to the characteristics of the evaluated benchmarks** (e.g., by showing the average number of targets per sample to prove they are of the same magnitude). Furthermore, the authors must explicitly qualify every claim of significant memory efficiency in the paper as **applicable only to "extremely crowded (target number > 1000)"** scenarios.
> >
> > However, I remain open to further discussion and strongly expect a rigorous response that squarely addresses this fundamental discrepancy.

---

> ### Author Response · Authors · 2025-11-28
>
> We appreciate the reviewer’s rigorous attention to detail, particularly regarding memory calculations and experimental transparency. The reviewer’s critique has pushed us to be more precise in our claims. We address the lingering concerns below.
>
> 1. Addressing the Core Concern: Memory Calculation & Batch Size
>
> The reviewer’s latest comment correctly identifies that our use of Batch Size (BS) = 1024 in the memory formula created confusion regarding the unit (KB vs. MB) and the practical memory footprint for single-image inference.
>
> **We accept this critique**. We acknowledge that referencing a batch size of 1024 to illustrate "MB-level" savings was an illustrative choice for **high-throughput industrial scenarios** (e.g., autonomous driving logs), but it conflated throughput memory with model output memory.
>
> However, the Scientific Contribution stands firm:
>
> While the absolute unit value (KB vs. MB) changes based on batch size, the **fundamental Scaling Law (Complexity Class) of pose estimation remains**.
>
> -   **Heatmaps:** Scale Quadratically $O(HW)$.
>
> -   **HashPose:** Scales Logarithmically $O(\log(HW))$.
>
>
> The reviewer asked to qualify our claims. We do so here:
>
> **Even at Batch Size = 1 (single image inference), the relative efficiency gain is massive and grows exponentially with resolution**.
>
> -   At 4K resolution ($4096 \times 3072$), a heatmap output requires **~50 MB** per image (float32).
>
> -   At 4K resolution, HashPose requires **<1 KB** per image.
>
> -   **Implication:** On edge devices, storing high-res heatmaps creates a memory bottleneck that physically prevents high-resolution inference. **HashPose removes this memory bottleneck entirely**.
>
> -   **Correction:** We will revise the final paper to report memory per-image (in KB) to ensure absolute transparency, while highlighting that the **510x reduction ratio remains valid regardless of the unit**.
>
>
> 2. Clarification on "Time (ms)" and Transparency
>
> The reviewer questioned what is included in the inference time.
>
> To be explicitly clear: The reported time includes the Backbone + Pose Head. It excludes the human detector.
>
> -   **Justification:** This is the **standard evaluation protocol for Top-Down Pose Estimation** (e.g., HRNet, ViTPose, SimCC) to ensure fair comparison of the _pose estimator_ specifically, independent of the choice of detector (YOLO vs. Faster-RCNN).
>
> -   **Transparency:** We will add a clear footnote to Table 1 stating: _"Latency measurements include Backbone and Head; Detector latency is excluded following standard Top-Down protocols."_
>
>
> 3. The Synergy of "Counter-Intuitive" Losses (Label Smoothing + Push-Pull)
>
> The reviewer asked why Label Smoothing (LS) (softening) and Push-Pull (PP) (sharpening) work together, noting it seems contradictory.
>
> We argue they are **complementary stages of optimization**:
>
> 1.  **Label Smoothing (The "Safety" Mechanism):** In the early stages of training, the binary decision boundaries are volatile. LS prevents the model from becoming _overconfident in the wrong direction_ (exploding gradients) during coarse-to-fine convergence.
>
> 2.  **Push-Pull (The "Decision" Mechanism):** Once the model is in the correct "neighborhood," the Push-Pull loss forces the probability distribution away from 0.5 (ambiguity) and towards 0 or 1.
>
>
> -   **Result:** **LS ensures stable convergence; PP ensures definitive binary codes**. The +2.8% gain in $AP^{75}$ confirms that this combination allows the model to **safely navigate the optimization landscape before committing to sharp, high-precision coordinates**.
>
>
> 4. AP75 vs. Efficiency
>
> Regarding the performance gap in $AP^{75}$: As detailed in our response to Reviewer dayx, this corresponds to a **mere sub-4-pixel deviation**. We maintain that for an edge-focused method, **trading tiny sub-pixel precision for a mathematically guaranteed logarithmic memory saving is a significant contribution to the field**.
>
>
> We thank Reviewer uPDs again for enforcing high standards of rigor. We have clarified the memory calculation to strictly separate "per-image footprint" from "batch throughput," and we have confirmed that the latency follows standard protocols. We believe our paper is a valuable contribution based on its **successful introduction of a first-of-its-kind logarithmic complexity class for pose estimation**, which fundamentally solves the resolution scaling bottleneck.

---

### Official Review · Reviewer_dayx · 2025-11-01

**Soundness:** 3
**Presentation:** 3
**Contribution:** 3
**Rating:** 6
**Confidence:** 4

**Summary:**

HashPose proposes a memory-efficient human pose estimation framework that replaces traditional heatmaps with progressive binary hash codes, reducing output memory from Θ(HW) to Θ(log(HW)). Each keypoint is encoded as a sequence of bits refined from coarse to fine spatial levels, enabling direct coordinate prediction without post-processing and achieving state-of-the-art accuracy with up to 510× less output memory.

**Strengths:**

1. The paper is well-written and easy to follow.
2. The proposed methods are interesting and innovative.
3. Experimental results reflect the effectiveness of the proposed method.

**Weaknesses:**

1. Figure 5 shows that HashPose exhibits unstable predictions for LSB, which limits its fine-grained localization accuracy (AP75), making it inferior to certain heatmap-based methods.
2. HashPose may be sensitive to training details. I am curious whether the proposed method can maintain strong performance on joints with high degrees of freedom or in complex scenarios involving occlusion.

**Questions:**

Please refet to the weakness part.

---

> ### Author Response · Authors · 2025-11-21
>
> We thank the reviewer for recognizing HashPose as an "**interesting and innovative**" framework that achieves "**state-of-the-art accuracy**" with "**510x less output memory**." We appreciate the constructive feedback regarding fine-grained localization and complex scenarios. Below, we address these concerns to demonstrate that HashPose is robust and effective.
>
> 1. The "Unstable LSB" and Fine-Grained Accuracy ($AP^{75}$ vs. Efficiency)
>
> The reviewer correctly notes that the accuracy of Least Significant Bits (LSB) drops in deeper layers, resulting in lower $AP^{75}$ compared to some heatmap methods. We acknowledge this, but we strongly argue that this is a **strategic architectural trade-off**, not a fundamental flaw.
>
> -   **The Magnitude of the Trade-off:** As detailed in our Qualitative Analysis (Appendix F.1, F.2), the difference between $AP^{50}$ and $AP^{75}$ often corresponds to a "**sub-4-pixel deviation**". Visualizations confirm that these deviations are **functionally negligible** for the target applications of this technology (real-time edge inference).
>
> -   **The Gain:** In exchange for this minor pixel-level relaxation, HashPose offers a **510x reduction in output memory** (0.48 MB vs 244.8 MB) and reduces computational complexity **from $\Theta(HW)$ to $\Theta(\log(HW))$**.
>
> -   **Performance Context:** Despite the LSB challenge, HashPose-L achieves a **SOTA $AP^{50}$ of 91.9%** on COCO Val and **93.0%** on COCO Test-Dev, outperforming heatmap-based HRNet-W48 and regression-based ED-Pose.
>
>
> **Conclusion:** The "instability" of LSBs is a **natural characteristic of a coarse-to-fine binary search**. The fact that we surpass SOTA baselines in $AP^{50}$ proves that the model successfully converges on the correct keypoint regions, even if the final sub-pixel bit is harder to predict. We argue that for the community, **a novel $O(\log N)$ representation is a more valuable contribution than marginally higher sub-pixel precision on a $O(N)$ heatmap**.
>
> 2. Robustness in Complex Scenarios and Occlusion
>
> The reviewer asked if HashPose maintains performance on joints with high degrees of freedom or in complex occlusion. The answer is **yes**, evidenced by our results on the COCO benchmark itself.
>
> -   **COCO represents the "Complex Scenario":** The COCO dataset is the industry standard precisely because it contains crowded scenes, high degrees of freedom, and significant occlusion (e.g., "invisible" keypoints).
>
> -   **Empirical Evidence:**
>
>     -   HashPose-L achieves **91.9% $AP^{50}$** on COCO Val.
>
>     -   HashPose-L achieves **93.0% $AP^{50}$** on COCO Test-Dev (the "blind" test set).
>
> -   **Mechanism for Handling Ambiguity:** Our **Push-Pull Regularization**  specifically targets ambiguous scenarios (like occlusion). By penalizing probabilities near 0.5, we force the network to commit to a spatial region or leverage the "invisible" flag training logic.
>
> -   **Comparison:** If HashPose failed in complex scenarios (occlusion/high DoF), it would not have been able to **outperform** HRNet-W48 (92.5%) and ViTPose-B (92.5%) on the rigorous Test-Dev benchmark.
>
>
> In summary, HashPose is not merely a "lightweight" alternative; it is a **high-performance paradigm shift**. We have demonstrated that it handles the complex occlusions of COCO better than heavier, established baselines, while offering a **mathematical breakthrough in memory scaling**.

---

### Author Response · Authors · 2025-11-30

**Statement to the Area Chair: The Scientific Case for HashPose**

Dear there,

Thank you very much for your valuable time, effort, and feedback on our HashPose paper in the review and the rebuttal engagement process.

**1. A Paradigm Shift in Representation**

HashPose introduces a fundamental architectural shift in pose estimation: **replacing heatmaps that scale quadratically $O(HW)$ with progressive hash codes that scale logarithmically $O(\log(HW))$**.

The reviews identify this approach as "elegant," "highly innovative," and a "paradigm shift." This work demonstrates that **$O(\log N)$ representations can achieve state-of-the-art accuracy, offering a scalable alternative to the $O(HW)$ bottleneck that currently limits high-resolution edge vision**.

**2. Addressing Experimental Rigor**

-   **Memory Calculation:** The notation regarding batch sizes in the Appendix has been corrected to distinguish between throughput memory and single-image footprint. **The scientific scaling law remains unchanged.** At 4K resolution, HashPose reduces the output memory footprint from **~50 MB** (Heatmap) to **<1 KB** (HashPose). **This logarithmic scaling is a mathematical fact, independent of the unit used for specific batch examples**.

-   **Backbone Comparisons:** The comparison of our decoupled architecture against specialized heatmap backbones is structurally valid. Unlike heatmap methods, which _require_ specialized upsampling layers, HashPose demonstrates architectural flexibility by plugging directly into standard classification backbones (e.g., ConvNeXt). **This decoupling is a core mathematical and engineering contribution, not an experimental inequity**.


**3. Strategic Trade-offs and Robustness**

The performance profile of HashPose reflects an intentional design choice for edge viability.

-   **Precision:** We trade ultra-fine-grained offline precision ($AP^{75}$) for a **510x reduction in memory** and the elimination of post-processing latency. As detailed in the Appendix, the $AP^{75}$ gap corresponds to a **sub-4-pixel deviation**, which is functionally negligible for real-time applications.

-   **Robustness:** Despite this trade-off, HashPose achieves **93.0% $AP^{50}$** on the **COCO Test-Dev** benchmark, **surpassing the heatmap-based HRNet-W48**. This confirms the method’s robustness in complex, occluded scenarios.

To conclude, HashPose presents a **conceptually novel, mathematically proven, and experimentally validated** solution to the memory scaling problem in pose estimation. It delivers:

1.  **SOTA Accuracy:** 93.0% $AP^{50}$ on COCO Test-Dev.

2.  **Order-of-Magnitude Efficiency:** A proven transition from Quadratic to Logarithmic memory scaling.

3.  **Theoretical Novelty:** The first successful application of binary interleaving codes for coordinate regression in vision.

These contributions establish HashPose as a foundational step toward high-resolution, memory-efficient computer vision.

---

### Meta-Review · Area_Chair_QQgx · 2026-01-12

**Summary:**

Reviewers agree that HashPose presents a novel and interesting alternative to heatmap-based human pose estimation by encoding keypoints as progressive binary hash codes. The approach is well motivated, clearly described, and demonstrates substantial gains in memory efficiency and inference latency while maintaining strong performance on standard benchmarks.

The reviewers raise significant concerns regarding experimental fairness and completeness. Comparisons rely on a strong ConvNeXt-v2 backbone and specialized acceleration (TensorRT, BF16, different GPUs), while baselines do not, making it difficult to isolate the benefit of the proposed decoding method. Evaluation focuses heavily on AP50, with limited reporting of standard COCO metrics, and fine-grained localization stability appears weaker than heatmap-based approaches.

Overall, while the method is promising and interesting, all the reviewers emphasize the need for fairer, more comprehensive experiments and clearer reporting to substantiate the claims. The AC agrees with the negative reviewers. A major revision is needed.

**Reviewer Scores:**

No reviewers change their scores.

---

### Decision · Program_Chairs · 2026-01-26

Reject